# Expression of a Large Single-Chain 13F6 Antibody with Binding Activity against Ebola Virus-Like Particles in a Plant System

**DOI:** 10.3390/ijms21197007

**Published:** 2020-09-23

**Authors:** Sohee Lim, Do-Sun Kim, Kisung Ko

**Affiliations:** 1Department of Medicine, College of Medicine, Chung-Ang University, Seoul 06974, Korea; annysh520@gmail.com; 2Vegetable Research Division, National Institute of Horticultural and Herbal Science, Rural Development Administration, Jeonju 55365, Korea; greenever@korea.kr

**Keywords:** Ebola, ZEBOV, 13F6, large single chain antibody, KDEL, plant-derived antibody

## Abstract

Pathogenic animal and human viruses present a growing and persistent threat to humans worldwide. Ebola virus (EBOV) causes zoonosis in humans. Here, two structurally different anti-Ebola 13F6 antibodies, recognizing the heavily glycosylated mucin-like domain (MLD) of the glycoprotein (GP), were expressed in transgenic *Nicotiana tabacum* plants and designed as inexpensive and effective diagnostic antibodies against Ebola virus disease (EVD). The first was anti-EBOV 13F6 full size antibody with heavy chain (HC) and light chain (LC) (monoclonal antibody, mAb 13F6-FULL), while the second was a large single-chain (LSC) antibody (mAb 13F6-LSC). mAb 13F6-LSC was constructed by linking the 13F6 LC variable region (VL) with the HC of mAb 13F6-FULL using a peptide linker and extended to the C-terminus using the endoplasmic reticulum (ER) retention motif KDEL. *Agrobacterium*-mediated plant transformation was employed to express the antibodies in *N. tabacum*. PCR, RT-PCR, and immunoblot analyses confirmed the gene insertion, transcription, and protein expression of these antibodies, respectively. The antibodies tagged with the KDEL motif displayed high-mannose type *N*-glycan structures and efficient binding to EBOV-like particles (VLPs). Thus, various forms of anti-EBOV plant-derived mAbs 13F6-FULL and LSC with efficient binding affinity to EBOV VLP can be produced in the plant system.

## 1. Introduction

Ebola virus disease (EVD), also known as Ebola hemorrhagic fever, is an acute, severe, and often fatal illness. The average mortality rate of EVD is approximately 50%, varying from 25% to 90% [1,2,3]. Since the discovery of EVD in 1976, its largest outbreak has been the 2014–2016 outbreak in West Africa [2,3], wherein the Ebola virus (EBOV) caused approximately 11,309 deaths in 28,645 cases [4,5]. The current ongoing 2018 outbreak in the eastern Democratic Republic of the Congo (DRC) caused the death of 2299 people among 3481 infected cases, and is still in progress, according to the World Health Organization (WHO) on 3 July 2020 [6]. Thus, EBOV is a global health concern and is under study by a widespread scientific collaboration.

The incubation period of EVD ranges from 2 to 21 days and the symptoms of EVD include fever, fatigue, headache, muscle pain, vomiting, diarrhea, and bleeding [3,7]. Because of its infectiousness and high mortality, it is important to rapidly diagnose, quarantine (separated from other people), and treat people infected with EBOV. The diagnostic tests of clinical laboratory settings include three basic categories. The first category is antigen testing to detect viral proteins, the second is molecular testing to detect viral RNA sequence, and the third is serologic testing to detect host antibodies generated against EBOV [8]. Since the EBOV levels in the blood typically increase within the early days of the symptoms, antigen and molecular diagnostics have proven very efficient and effective for acute diagnosis [8,9].

The 2014–2016 outbreak and the current outbreak in the DRC are caused by the Zaire Ebolavirus species (ZEBOV), which is the most virulent species among the Ebola genus according to the WHO [3,10,11]. ZEBOV has trimeric glycoproteins (GP) forming the structure of the surface EBOV and is known as one of the crucial factors in determining its differential virulence [5,10,12]. The functions of this GP are cellular attachment to the host cells leading to virus penetration, membrane fusion, and endosomal entry [5,8,13,14]. In addition, anti-GP antibodies display a protective effect against EVD as seen from the lethal challenge animal models [12,15]. This GP is a key target for the design of an Ebola vaccine, therapeutic approaches, and diagnostic targets [8,16].

The murine/human chimeric 13F6 antibody is one of four highly protective antibodies that recognize the GP protein of ZEBOV [17,18]. The monoclonal antibody (mAb) 13F6 recognizes the amino acid residues 405–413 that form the heavily-glycosylated mucin-like domain (MLD) of ZEBOV GP and has been shown to possess a therapeutic protective efficacy in a lethal challenge mouse model [17]. In addition, the 13F6 antibody is one component of the mAb-based EBOV immunoprotectant MB-003 (13C6, 13F6, and 6D8) [19,20]. MB-003 was shown to protect against lethal challenge in EBOV-infected *Rhesus macaque* model at 24 or 48 h after virus exposure [20,21].

The EBOV antigen detection assay is an accurate and rapid antibody-based diagnostic test. To make this diagnostic test cost-effective and easily accessible, low cost production of antibodies with a strong binding affinity against the EBOV antigens is crucial. One option to ensure cost-effective production of antibodies is the use of a plant expression system without pathogenic animal contaminants, high-priced manufacturing process, and expensive culture media [22,23,24,25,26]. The advantages of plant expression systems include inexpensive inputs (sunlight, water, and nutrients), the ease of large-scale production by increasing the area located in the seed, and low technology harvest compared with mammalian expression systems [22,27,28,29]. Moreover, plant expression systems are suitable for post-translational modifications of proteins, including the *N*-glycosylation required for biological activity, similar to mammalian expression systems [25,28,30,31]. The structure of the antibody should be properly modified for efficient antibody production and its assembly in the plant expression system. The large single-chain (LSC) antibody structure has several advantages. The light chain (LC) variable region (VL) and heavy chain (HC) coding sequences are genetically linked with the help of a peptide linker in a single transcript. Thus, there is no need to balance expressing proportion of the LC and HC. In addition, The LSC antibody required only one promoter to express in plant [32]. This allows LSC antibody to be more quickly produced at lower costs than full size antibody [33,34]. The Fc region of LSC form facilitates efficient purification by affinity chromatography using protein G or protein A beads. Expression of two different targeting full size antibodies in a single plant could provoke misassembly of their own heavy and light chains, consequently resulting in chimeric antibodies with loss activities [35]. Thus, in the present study, a large single-chain (LSC) type antibody, which can be expressed in a single plant cell without the chimerism was constructed by using a single promoter and terminator rather than two promoters and terminators for the formation of the heavy and light chains of a full-size antibody.

Although 13F6 has no neutralizing activity on EBOV, mAb 13F6 is one of the highly protective antibodies that recognize the GP protein of ZEBOV. In addition, the mAb 13F6 has several important characters, such as a specific MLD binding activity without interfering interaction of mAb 12B5 to the MLD, which could provide important information for development of anti-EBOV mAb cocktails including advantage for quick EBOV detection [36]. Thus, focusing on the production of an EBOV diagnostic antibody in this study, the full type 13F6 monoclonal antibody (mAb 13F6-FULL) and an LSC type 13F6 monoclonal antibody (mAb 13F6-LSC) were expressed in transgenic *Nicotiana tabacum* and their in vitro efficacy was confirmed. Our data suggested that various forms of the anti-EBOV antibodies were well expressed in and purified from the transgenic *N. tabacum* plant. Moreover, the mAb 13F6-LSC displayed a better binding to EBOV-like particles (VLPs) than that displayed by the mAb 13F6-FULL.

## 2. Results

### 2.1. Expression of mAb 13F6-FULL and mAb 13F6-LSC in Transgenic N. tabacum Plants

Transgenic *N. tabacum* lines expressing anti-EBOV mAb 13F6-FULL and mAb 13F6-LSC were generated by *Agrobacterium*-mediated transformation (Figure 1). The modification of the antibody structure was achieved by linking the mAb 13F6 VL with the HC of mAb 13F6-FULL using a peptide linker. The glycan structures were modified by employing the endoplasmic reticulum (ER) retention motif KDEL to yield a high-mannose type *N*-glycan structure (Figure 1A). Gene insertion and mRNA expression of these two antibodies were analyzed by PCR and RT-PCR analyses (Figure 2) that were performed in three randomly-selected transgenic lines from *N. tabacum*. PCR analysis confirmed the presence of the HC, LC, and LSC genes in the genomic DNA (gDNA) isolated from the 13F6-FULL and 13F6-LSC transgenic plants since the amplified HC and LC fragments of mAb 13F6-FULL and the LSC fragment of mAb 13F6-LSC were detected in all samples. No bands were detected in the non-transgenic *N. tabacum* plant (NT). Whole-gel images of Electrophoresis were provided in Appendix A. Immunoblot analysis was employed to evaluate the protein expression of the HC and LC of mAb 13F6-FULL and that of the LSC of mAb 13F6-LSC on three lines of each 13F6-FULL and 13F6-LSC transgenic plant in the same transgenic lines as described above (Figure 3). The HC (50.9 kDa) and LSC bands (67.2 kDa) were identified in the in vitro extracts of the total soluble protein from the leaves of the transgenic *N. tabacum* plant by anti-human immunoglobulin G (IgG), Fcγ fragment specific antibody. The LC bands (24.3 kDa) were detected by the anti-human IgG, F(ab′)_2_ fragment specific antibody. The mAb 13F6-FULL showed higher levels of protein expression than that of the mAb 13F6-LSC. 

### 2.2. Purification of mAb 13F6-FULL and 13F6-LSC Antibodies from the Transgenic N. tabacum Plant

Transgenic plant lines expressing high levels of mAb 13F6-FULL and 13F6-LSC were identified by the immunoblot assay. Each selected plant line was transferred to soil pots and grown in a greenhouse for mass production (Figure 1B). The mAb 13F6-FULL and mAb 13F6-LSC were purified from fresh harvested leaves (350 g). The yields of purified mAb 13F6-FULL and mAb 13F6-LSC were 24 mg and 7 mg per kilogram (kg) of harvested *N. tabacum* leaves, respectively. SDS-PAGE with Coomassie brilliant blue staining was performed for each eluted fraction sample (Figure 4) for the confirmation of the purified antibodies. The expected HC (black arrow head) and LC (white arrow head) bands of mAb 13F6-FULL were detected at ~50 kDa and ~25 kDa in the eluted fractions. The expected LSC band (gray arrow head) of mAb 13F6-LSC was detected at ~67 kDa in the eluted fractions. As seen from SDS-PAGE, mAb 13F6-FULL was remarkably more pure than the mAb 13F6-LSC.

### 2.3. N-glycan Profile Analysis of mAb 13F6-FULL and mAb 13F6-LSC Using HPLC

HPLC was performed to analyze the *N*-glycan profiles released from mAb 13F-C, mAb 13F6-FULL and mAb 13F6-LSC (Figure 5). mAb 13F6-FULL and mAb 13F6-LSC showed oligo-mannose-type glycan structures. They had a similar structure consisting of mostly 6–9 mannose glycans. mAb 13F6-FULL (Figure 5 middle) had a significantly higher 8-mannose-type glycan structure than that of mAb 13F6-LSC (Figure 5 bottom). However, in mAb 13F6-C, mainly *N*-glycans with β1,2-linked xylose and α1,3-linked fucose residues were observed without an oligo-mannose-type glycan structure (Figure 5 top).

### 2.4. Binding of mAb 13F6-FULL and mAb 13F6-LSC to Ebola VLP

The binding activity of mAb 13F6-FULL and mAb 13F6-LSC to Ebola VLP was confirmed by the immunoblot assay (Figure 6). Ebola VLPs (amounts: 50 and 100 ng) and 1× phosphate-buffer saline (PBS) were loaded on a 10% SDS-PAGE gel and treated with purified mAb 13F6-FULL and mAb 13F6-LSC from the plant. This antibody is a constitutive antibody of the EBOV immunoprotectant MB-003 and was known to display protective action in mouse and *Rhesus macaques* lethal challenge models [17,19,20,21]. All antibodies interacted with VLP and three bands of Ebola VLPs were seen in each lane except for the 1× PBS (-) lane. The binding activity of mAb 13F6-FULL and mAb 13F6-LSC to Ebola VLP was measured by the ELISA analysis (Figure 7). The 96-well MaxiSorp Nunc-immuno plates were coated with 1 µg of Ebola VLPs and then treated with purified mAb 13F6-FULL and mAb 13F6-LSC. The commercial anti-ZEBOV GP mAb 13F6 (mAb 13F6-C) and 1× PBS were used as a positive and a negative control, respectively. To confirm the interaction between the mAbs 13F6 and the Ebola VLPs, peroxidase-conjugated AffiniPure goat anti-human IgG Fcγ fragment specific antibody was used as a secondary antibody. As shown in the immunoblot data for binding activity (Figure 6), all three antibodies detected the VLP protein band. In the ELISA analysis, the mAb 13F6-LSC displayed a higher binding activity than the other antibodies at the same concentration. mAb 13F6-FULL displayed a better binding than the mAb 13F6-C at the highest concentration.

### 2.5. Comparison of the Binding Affinity of mAb 13F6-C, mAb 13F6-FULL, and mAb 13F6-LSC to the Ebola VLP Using Surface Plasmon Resonance (SPR)

The binding affinity of mAb 13F6-C, mAb 13F6-FULL, and mAb 13F6-LSC to the Ebola VLP was measured using SPR (Figure 8). The Ebola VLP was immobilized onto a PEG sensor chip and a series of different concentrations of mAb 13F6-FULL, mAb 13F6-LSC, and the commercial anti-ZEBOV GP mAb 13F6 (mAb 13F6-C as a positive control) were applied and the binding kinetics examined. The association and dissociation rates for mAb 13F6-C were 2.14 ± 0.02 × 10^5^ M^−1^s^−1^ and 2.61 ± 0.01 × 10^−3^ s^−1^, respectively (Figure 8A,D). The binding affinity (K_D_) was calculated as 12.2 ± 0.1 nM (Figure 8D). The association and dissociation rates for mAb 13F6-FULL were 0.81 ± 0.004 × 10^5^ M^−1^s^−1^ and 0.75 ± 0.005 × 10^−3^ s^−1^, respectively. The binding affinity (K_D_) was calculated as 9.26 ± 0.04 nM (Figure 8B,D). The association and dissociation rates for mAb 13F6-LSC were 0.251 ± 0.002 × 10^5^ M^−1^s^−1^ and 0.251 ± 0.002 × 10^−3^ s^−1^, respectively (Figure 8C,D). The binding affinity (K_D_) was calculated as 8.56 ± 0.06 nM (Figure 8D). Therefore, the association and dissociation rates of mAb 13F6-FULL and mAb 13F6-LSC were lower than that of mAb 13F6-C.

## 3. Discussion

Infectious diseases of foreign origin have shown an increasing trend worldwide. They originate from many factors, including the movement of biological life by international travel and the globalization of trade and commerce [37,38,39]. These infectious diseases are endemic, epidemic, and pandemic in nature and include coronavirus disease (COVID-19), severe acute respiratory syndrome (SARS), and the Middle East respiratory syndrome (MERS) [40]. Although studying these infectious diseases is crucial for ensuring effective treatment, early and accurate diagnosis of these infectious diseases is equally essential. An early isolation of patients by a precise diagnosis is critical for preventing the spread of any infectious disease [41]. Moreover, an early treatment regime of patients with EVD can tremendously improve their chances of survival [42]. Therefore, early and rapid diagnosis has an important role in determining the success of any intervention strategy during outbreaks [41].

EBOV, a member of the *Filoviridae* family, causes a lethal hemorrhagic fever and is a global health concern [11]. Rapid and accurate diagnosis of EVD plays a critical role in an outbreak response and can improve the chances of survival in patients with EVD [42]. Among the diagnostic tests, EBOV antigen detection test by chromatographic lateral flow immunoassay takes an average of 15–30 min to show the results [8]. This test is an antibody-based assay and targets the GP of the EBOV for diagnosis [8,16]. In the current study, we produced anti-EBOV mAb 13F6-FULL and mAb 13F6-LSC in transgenic plants. Their binding affinity to Ebola VLPs was investigated to determine whether the LSC form of the anti-EBOV mAb can be expressed and properly assembled for use as a diagnostic antibody.

The pBIN mAb 13F6-FULL plant expression vector was constructed under the control of the Pin2 and 35S promoters for LC and HC, respectively, and the pBI mAb 13F6-LSC vector was constructed under the control of the 35S promoter for *Agrobacterium*-mediated transformation. In general, all antibodies comprise the HC and LC. Thus, two promoters are required to express and assemble a complete full-size antibody. A well-balanced proportion of the LC and HC has a beneficial effect on the maximum production of an antibody [43]. However, the production of an entire antibody under the control of two promoters can give different expression rates for each of the chains and may result in the production of free LC or HC [44]. However, in the LSC-type antibody, the C-terminus of the VL was linked to the HC. Thus, only one promoter was required to express and assemble the LSC antibody and free LC or HC were not produced.

In this study, both gene insertion and mRNA expression of mAb 13F6-FULL and mAb 13F6-LSC were confirmed by PCR and RT-PCR analyses from randomly selected transgenic lines. The antibodies displayed different levels of gene insertion and mRNA expression. Immunoblot analysis confirmed the protein expression of each antibody in the transgenic lines. Although the protein expression of the antibodies showed some variation, the mAb 13F6-FULL and mAb 13F6-LSC were strongly and stably expressed in each transgenic plant. From the immunoblot results in Figure 3, the HC and LC bands were slightly higher than that of the commercial 13F6 antibody. This is probably due to the difference in glycan structure as shown in Figure 5. The glycosylation pattern of mAb 13F6-FULL is high mannose, whereas the glycosylation pattern of mAb 13F6-C was plant specific glycan structure with β1,2-linked xylose and α1,3-linked fucose residues. Thus, the molecular weight of mAb 13F6-FULL with high mannose glycan structure seems to have been shifted compared to the control 13F6-C. In previous studies, the molecular weight of recombinant GA733-2-Fc-KDEL proteins with high-mannose type *N*-glycans was higher than recombinant GA733-2-Fc with plant-derived *N*-glycan [45,46].

After production of the entire transgenic plant biomass in the greenhouse, the antibodies were purified from 350 g of the harvested leaves using a protein A affinity resin, which captured the Fc region of the antibody. Consequently, the amount of the purified mAb 13F6-FULL (24 mg/kg) was almost three times higher than that of the mAb 13F6-LSC (7 mg/kg). The antibodies were successfully purified for the *N*-glycan structure analysis and in vitro binding affinity analysis to Ebola VLPs. The higher purification efficiency of both antibodies might be due to the effect of the C-terminal-fused ER retention motif KDEL. Previous studies prove that mAbs were not retained in the ER in the absence of an ER retention motif [25,26,47,48,49]. They indicated that a protein fused with the KDEL motif at the C-terminal end is retained in the ER or is returned to the ER and its production levels are also improved [2,50,51,52]. The ER is a vast subcellular location for the accumulation of proteins that do not require the proteolytic process found in the secretory pathway of intracellular and extracellular spaces [23,24,25,26,28,53,54]. The figure of Immunoblot analysis to confirm the purified fraction of mAb 13F6-FULL from transgenic *N. tabacum* plants for Figure 4A was included as a Appendix A. As a result of immunoblot analysis confirming of the purified fraction and column through (C.T) of mAb 13F6-FULL, bands of HC and LC were not detected in C.T lane. In general, the bands, which look like HC and LC in the C.T lane from the SDS-PAGE result in Figure 4A, appear to be subunits of rubisco (D-ribulose 1,5-bisphosphate carboxylase/oxygenase) protein, which is the most abundant protein in plants [55]. In previous studies, in the results of SDS-PAGE and western blot of LSC-type antibodies purified from plants with the protein A affinity column, the rubisco protein was confirmed at the position of 50 kDa [32]. The HPLC analysis confirmed that both purified antibodies had oligo-mannose-type glycan structures. These *N*-glycosylation analysis results indicated that the ER retention motif KDEL efficiently retained the antibodies in the ER and yielded antibodies with an oligo-mannose-type glycan [23,25,28,47]. In previous studies, the anti-cancer antibodies with and without KDEL expressed in plant showed oligomannose type and plant-specific glycan profiles, respectively, whereas the antibodies expressed in mammalian cells had glycan structures harboring mammalian specific glycan resides, such as α1,6-linked fucose, β1,2-linked galactose, and α2,6-sialic acid [22,26,32,56].

An immunoblot assay was performed to confirm the binding activity of the purified mAb 13F6-FULL and mAb 13F6-LSC to the Ebola VLP antigen. Both antibodies interacted with Ebola VLPs in a manner similar to that of the commercial anti-ZEBOV GP mAb 13F6. This constitutive antibody, MB-003, is one of the candidates for EVD treatment and has proved effective in mouse and *Rhesus macaques* lethal challenge models [17,19,20,21]. A pattern of three bands of Ebola VLPs was seen in each lane except in the negative control (1× PBS). In comparison with mAb 13F6-FULL, mAb 13F6-LSC had a slightly higher binding activity to the VLP. The binding activity of mAb 13F6-FULL and mAb 13F6-LSC to the Ebola VLP was reconfirmed by ELISA analysis. The binding of the mAb 13F6-LSC to Ebola VLPs was significantly stronger than that of the other antibodies. The mAb 13F6-FULL had a slightly higher binding activity than that of the commercial mAb 13F6-C. These data indicated that the mAb 13F6-FULL and mAb 13F6-LSC were produced, assembled, and purified in a stable form from the plant. Moreover, they displayed a more efficient binding ability for Ebola VLPs than that of the commercial anti-ZEBOV GP mAb 13F6.

Furthermore, an SPR analysis was conducted to confirm the binding affinity between the purified mAb and Ebola VLPs. Different concentrations of the mAb 13F6-FULL, mAb 13F6-LSC, and the mAb 13F6-C were applied to the Ebola VLP-immobilized sensor chip. The positive control, mAb 13F6-C, showed a higher association and dissociation rates than those of the mAb 13F6-FULL and mAb 13F6-LSC. The association and dissociation of mAb 13F6-FULL and mAb 13F6-LSC was slower than that of mAb 13F6-C. From the values of the binding affinity (K_D_), the purified mAb 13F6-FULL and mAb 13F6-LSC showed a higher binding affinity than that of the positive control. This was true especially for the mAb 13F6-LSC antibody that displayed a slightly stronger binding affinity than that of mAb 13F6-FULL antibody. In this study, the flexible glycine-rich-sequences as a linker was used to fuse VL to HC variable region (VH) in mAb 13F6-LSC, which could give more flexibility for better affinity to the antigen than the commercial full size mAb [57]. This higher binding activity can increase the sensitivity and specificity of a diagnosis. Therefore, mAb 13F6-LSC antibody is capable of acting as a diagnostic antibody in a manner better than the currently existing mAb 13F6 antibody. Crispin et al. [58] have reported that the oligo-mannose glycan structures of the antibody Fc region cause a 2 Å shifting of the Fc region, thereby the hinge angle of Cγ2−Cγ3 increases by 10° and 6.6° in each HC of the antibody, respectively. This shifting alters the structure of the antibody and increases the affinity with the Fc receptor that further enhances the effector function [58]. The enhanced binding affinity of the purified mAb 13F6-FULL and mAb 13F6-LSC could be due to alteration to the oligo-mannose glycan structure.

In this study, we revealed that mAb 13F6-FULL and mAb 13F6-LSC could be produced, assembled well, and purified from transgenic *N. tabacum* plants and display binding activity to Ebola VLPs similar to that of a commercial anti-ZEBOV GP mAb 13F6. In a previous study, the mAb 13F6-C, which was used as a positive control, has already been shown to be effective in binding to EBOV and in the mouse and *Rhesus macaques* lethal challenge models [18,20,59]. We found the binding affinity of mAb 13F6-LSC to be better than that of mAb 13F6-C. These results suggest that the biological activity of mAb 13F6-LSC antibody could be as efficient as that of mAb 13F6-C. However, further studies are needed because we have not yet identified the efficacy of the mAb 13F6-LSC as a component of any diagnostic test in patient samples of EVD. Moreover, it is not clear whether it can be reproduced under in vivo conditions and in samples of EVD patients. The mAb 13F6 does not have a neutralizing capacity to make it available as a therapeutic antibody. Nevertheless, the mAb 13F6-LSC with an enhanced binding affinity can be applied in the development of therapeutics for EVD. Taken together, this efficient and inexpensive plant-derived 13F6-LSC antibody production should be useful for advances in diagnostic testing in the current and future Ebola outbreak epidemics.

## 4. Materials and Methods

### 4.1. Construction of Expression Vectors for Plant Transformation

The synthetic DNA sequence encoding the mAb 13F6-FULL variable region (13F6 murine VH, GenBank accession no. JN374688 and 13F6 murine VL, GenBank accession no. JN374689) was fused with the human IgG1 [48] and lambda chain constant regions, respectively [18]. To construct the mAb 13F6-LSC, the synthetic DNA sequence encoding the mAb 13F6 VL was linked with the HC of mAb 13F6-FULL with the help of a peptide linker (Gly-Gly-Gly-Gly-Ser)_3_ as described in Mayfield et al. [60]. The mAb 13F6-FULL and mAb 13F6-LSC were modified by *N*-terminal extension with 30 amino acids using the plant ER signal peptide as described in Lu et al. [24] and modified at the C-terminal end with an ER retention motif (KDEL: Lys-Asp-Glu-Leu) as described in So et al. [61]. The LC and HC of mAb 13F6-FULL were subcloned under the control of the Pin2 promoter and the enhanced duplicated 35S promoter (E/35S-P) in the plant expression vector pBINPLUS, respectively. The mAb 13F6-LSC sequences were subcloned under the control of the E/35S-P in the plant expression vector pBI121 (Figure 1A).

### 4.2. Generation of Transgenic Plants and Growth Conditions

Plant transformation was performed with *A. tumefaciens* strain *LBA4404* as described by Park et al. [48] (Figure 1B). In vitro transgenic plants were grown in a growth chamber under controlled environmental conditions (130 μmol photons m^−2^ s^−1^ at 23 °C with a 70% relative humidity and a 16-h light/8-h dark cycle). For harvesting the biomass, the in vitro transgenic plants were transferred into soil pots and grown in a greenhouse.

### 4.3. Plant Genomic DNA, RNA Isolation, PCR, and RT-PCR Analyses

Plant genomic DNA was isolated from transgenic tobacco leaf tissues with the Genomic DNA Extraction Kit Mini (Plant) following the manufacturer’s instructions (RBC Bioscience, Taipei, Taiwan). Total plant RNA was isolated from the transgenic leaf tissue following the TRIzol RNA isolation protocol [62]. cDNA synthesis was performed using the QuantiTect Reverse Transcription Kit following the manufacturer’s instructions (Qiagen, Hilden, Germany). To confirm the gene insertion and expression of the mRNA, PCR and RT-PCR amplification of the plant 2- and synthetic cDNA was conducted using the following primer pairs: for 13F6-FULL HC: forward primer 5′-TGC CCT CCA GCA GCT TGG-3′, reverse primer 5′-GGT GGG CAT GTG TGA GTT TTG TC-3′; for 13F6-FULL LC: forward primer 5′-GGA TCC ATG GCT ACT CAA CGA AG-3′, reverse primer 5′-CTG CAG CTA TGA ACA TTC TGT AGG G-3′; for 13F6-LSC: forward primer 5′-CAA GGG CCC ATC GGT CTT C-3′, reverse primer 5′-CTG GTC AGG GCG CCT GAG T-3′; for NtEF-1α: forward primer 5′-GCT GCT GAG ATG CAC AAG CGG T-3′, reverse primer 5′-CAG GAC ACG ACA GGC ACG GGA-3′. The PCR and RT-PCR were programmed as follows: 30 cycles at 94 °C for 20 s, 56–59 °C for 30 s, and 72 °C for 30 s using the Maxime PCR PreMix Kit (iNtRON Biotechnology, Seoul, Korea). The relative expression level of each gene was normalized by using the *EF-1α* as the internal control. The positive controls were pBIN 13F6-FULL and pBI 13F6-LSC in *DH5α* and the negative controls were the gDNA and RNA isolated from non-transgenic plants.

### 4.4. SDS-PAGE and Immunoblot Analysis

To confirm the protein expression, 100 mg of fresh leaf samples from each in vitro transgenic plant were homogenized in three volumes of 1× PBS to extract the total soluble proteins. For SDS-PAGE, 20 µL of the prepared samples were mixed with 5× protein loading buffer (1% bromophenol blue, 5% 2-mercaptoethanol, 10% SDS, 50% glycerol, 1 M Tris-HCl) and boiled for approximately 5 min. The samples were loaded on a 10% SDS-PAGE gel and transferred to a nitrocellulose membrane (Merck Millipore, Burlington, NY, USA). For blocking, the membranes were incubated with 5% skimmed milk (Sigma-Aldrich, St. Louis, MO, USA) in 1× TBS plus 0.5% (*v*/*v*) Tween 20 (1× TBS-T). The blot membranes were incubated for 2 h at room temperature (RT) with peroxidase-conjugated AffiniPure goat anti-human IgG, Fcγ fragment specific antibody #109-035-008 (1:4000) (Jackson ImmunoResearch Laboratories, West Grove, PA, USA) and peroxidase-conjugated AffiniPure goat anti-human IgG, F(ab′)_2_ fragment specific antibody (1:4000) (Jackson ImmunoResearch Laboratories, West Grove, Chester County, PA, USA) for recognition of the HC and LC of mAb 13F6, respectively. The non-transgenic plant and the commercial anti-ZEBOV GP plant-derived mAb 13F6 (IBT BioServices, Rockville, MD, USA) were used as the negative control and positive control, respectively. To confirm the binding activity of plant-derived antibodies to VLPs (IBT BioServices, Rockville, MD, USA), 100 ng and 50 ng of VLP were loaded on a 10% SDS-PAGE gel and transferred to a nitrocellulose membrane. The membranes were incubated with mAb 13F6-FULL, mAb 13F6-LSC, and the commercial anti-ZEBOV GP mAb 13F6 (2 µg/mL) (IBT BioServices, Rockville, MD, USA) as the primary antibody for 2 h at RT. Subsequently, the membranes were incubated with peroxidase-conjugated AffiniPure goat anti-human IgG, Fcγ fragment specific antibody (1:4000) as the secondary antibody for 2 h at RT. The negative control was 1× PBS. The protein bands were visualized by exposing the membrane to an X-ray film (Fuji, Tokyo, Japan) using Clarity Western Enhanced chemiluminescence (ECL) Substrate (Bio-Rad, Hercules, CA, USA).

### 4.5. Purification and Dialysis of mAb 13F6-FULL and mAb 13F6-LSC

The mAb 13F6-FULL and mAb 13F6-LSC were purified from fresh leaves as described in a previous study [63]. Fresh harvested leaves weighing 350 g were homogenized in 1050 mL of pre-chilled extraction buffer (15 mM EDTA, 37.5 mM Tris-HCl, 50 mM NaCl, 75 mM sodium citrate, and 0.2% sodium thiosulfate, pH 7.4) and centrifuged at 9000× *g* for 30 min at 4 °C. The supernatant solution was filtered by a Miracloth (Biosciences, La Jolla, CA, USA) and its pH was adjusted to 5.1 by the addition of acetic acid (pH 2.4) for the removal of chloroplast. The solution was centrifuged at 10,000× *g* for 30 min at 4 °C. The supernatant solution was filtered through a Miracloth and its pH was adjusted to 7.0 by the addition of 3 M Tris-HCl solution; ammonium sulfate was added to 8% of the solution volume at 4 °C and incubated for 2 h. This incubated solution was centrifuged at 9000× *g* for 30 min at 4 °C and the supernatant solution was again filtered using a Miracloth. Ammonium sulfate was added to 22.6% of the solution volume and it was incubated overnight at 4 °C. Subsequently, the solution was centrifuged at 9000× *g* for 30 min at 4 °C and the pellet was resuspended in 1/10th the starting volume of the pre-chilled extraction buffer. This solution was centrifuged at 10,000× *g* for 30 min at 4 °C and filtered by a Miracloth and a 0.45-μm filter (Merck Millipore, Burlington, NY, USA). mAb 13F6-FULL and mAb 13F6-LSC were purified by protein A Sepharose 4 Fast Flow column (GE Healthcare, Chicago, IL, USA) following the manufacturer’s instructions. The purified antibodies were dialyzed with 1× PBS (pH 7.4) using a Por 2 RC Dialysis Membrane Tubing, MWCO 12,000 to 14,000 Dalton (Spectrum Chemical, New Brunswick, NB, USA). The protein concentration was determined by NanoDrop spectrophotometer (BioTek, Winooski, VT, USA) and SDS-PAGE analyses. The purified proteins were stored at −80 °C until further use.

### 4.6. HPLC Analysis

The dialyzed mAb 13F6-C, mAb 13F6-FULL, and mAb 13F6-LSC (100 μg each) were treated with pepsin at 37 °C for 24 h to digest the protein into glycopeptides. The digested glycopeptide samples were passed through a Sep-Pak C18 1 cc Vac Cartridge column (Waters Corporation, Milford, CT, USA) and washed with 5% acetic acid to remove impurities such as free glycans and salt. The fraction was eluted in a series of solutions with 20 and 40% isopropanol based on 5% acetic acid. After collecting the eluted fractions in one tube, the samples were dried in a rotary vacuum evaporator and incubated overnight at 37 °C with PNGase A (Roche, Basel, Switzerland) to release *N*-glycans. For purification of the glycans, the samples were passed through Carbograph SPE Columns (Alltech, Nicholasville, KY, USA) and dried in a rotary vacuum evaporator. The purified glycans were labeled with 2-Aminobenzamide (2-AB) for detection in HPLC as described in a previous study [28]. The labeled samples were diluted with 1 mL of 96% (*v*/*v*) acetonitrile in HPLC-grade water and passed through a Bond Elut CN-E cartridge (Agilent Technologies, Santa Clara, CA, USA) following the manufacturer’s instructions; this exercise helped to remove the excess labeling reagent. The purified 2-AB-labeled samples were separated on a TSKgel Amide-80 column (5 μm, 4.6 mm × 250 mm, Tosoh Bioscience, Tokyo, Japan) using the HPLC system with a 330 nm excitation and a 425 nm emission fluorescence detector. Separations were achieved at a flow rate of 1 mL/min using a mixture of 100% acetonitrile (solvent A) and 50 mM ammonium formate, pH 4.4 (solvent B). After the column was equilibrated with 30% solvent B for 10 min, the sample was injected. Elution was achieved with a linear gradient of 45% solvent B for 50 min.

### 4.7. ELISA Analysis

The 96-well MaxiSorp Nunc-immuno plates (Sigma-Aldrich, St. Louis, MO, USA) were coated with 1 µg per well of Ebola VLPs diluted in 1× DPBS (Welgene, Gyeongsan, Korea) and incubated overnight at 4 °C. The plates were washed with 200 μL of 1× DPBS plus 0.05% (*v*/*v*) Tween 20 (1× DPBS-T) per well for four times and blocked with 3% BSA (bioWORLD, Dublin, OH, USA) in 1× DPBS-T for 2 h at RT. After washing, 1.5625 ng to 25 ng of the mAb 13F6-FULL, mAb 13F6-LSC, and the commercial anti-ZEBOV GP mAb 13F6 (IBT BioServices, Rockville, MD, USA) were incubated in 1X DPBS-T for 2 h at 37 °C as the primary antibody. After washing, the plates were treated with peroxidase-conjugated AffiniPure goat anti-human IgG, Fcγ fragment specific antibody (dilution of 1:4000) and incubated for 2 h at RT. After another wash with 1× DPBS-T, the ELISA plates were detected using 100 μL per well of the SureBlue TMB Microwell Peroxidase Substrate and the TMB stop solution (SeraCare, Milford, CT, USA). The absorbance was measured at 450 nm using an Epoch microplate spectrophotometer (BioTek, Winooski, VT, USA).

### 4.8. SPR Analysis

The SPR analysis was conducted at 25 °C to confirm the binding of mAb 13F6-FULL, mAb 13F6-LSC, and commercial anti-ZEBOV GP mAb 13F6 (IBT BioServices, Rockville, MD, USA) to the Ebola VLP (IBT BioServices, Rockville, MD, USA) by the Reichert SR7500DC SPR spectrometry system (Reichert Analytical Instrument, Depew, NY, USA). The VLP in 20 mM sodium acetate buffer, pH 5.5 was immobilized using free amine coupling by activating the reactive succinimide ester surface on a PEG Chip (Reichert Analytical Instruments, Depew, NY, USA) through a mixture of 0.1 M 1-ethyl-3-(3-dimethylaminopropyl) carbodiimide hydrochloride and 0.05 M *N*-hydroxysuccinimide (Sigma-Aldrich, St. Louis, MO, USA) and stabilized with 1 M ethanolamine, pH 8.5 at a flow rate of 20 μL/min. To evaluate the binding affinity of mAb 13F6-FULL (3.125–200 nM) or mAb 13F6-LSC (1.5625–200 nM) or commercial anti-ZEBOV GP mAb 13F6 (1.5625–50 nM) to Ebola VLP, different concentrations of each antibody in running buffer (0.01 M phosphate buffer, 2.7 mM KCl, 0.137 M NaCl, 5% DMSO, pH 7.4 after 10 times dilution) were injected over the VLP-sensor chip (approximately 1530 RU) at a flow rate of 30 μL/min for 4 min for association and for 5 min for dissociation. The surface of the VLP-sensor chip was regenerated using 1–50 mM NaOH at a flow rate of 30 μL/min for 30–60 s. The kinetic constant values calculated using a simple 1:1 interaction model and the sensorgrams were fit using the Scrubber2 analysis program (BioLogic Software, Campbell, Canberra, Australia).

## Figures and Tables

**Figure 1 ijms-21-07007-f001:**
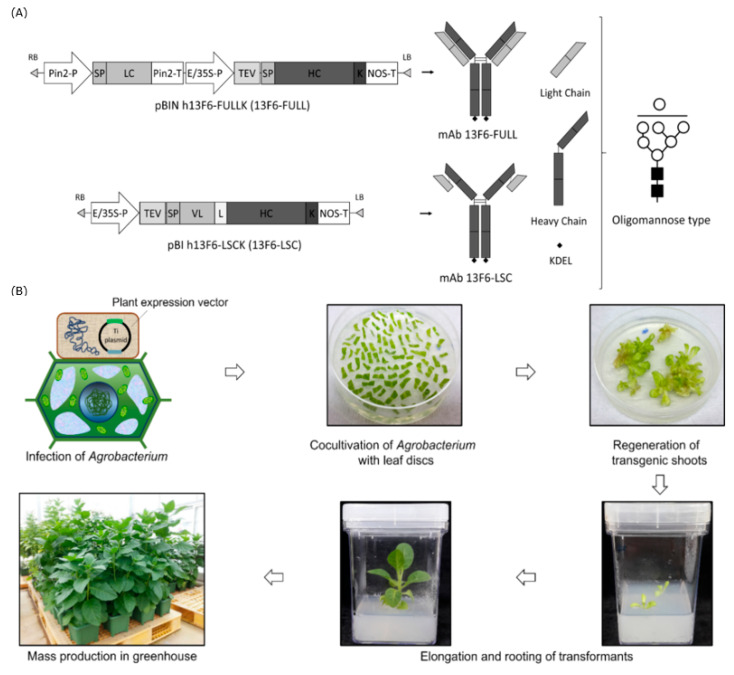
Schematic representation of the plant expression cassette and Agrobacterium-mediated plant transformation for the production of transgenic plants expressing 13F6 antibodies. (**A**) Plant expression cassettes of monoclonal antibody (mAb) 13F6-FULL and mAb 13F6-large single-chain (LSC) in plant expression vectors for plant transformation with expected protein and glycan structures. Key: Pin2-P, promoter of potato Pin2 gene; SP, signal peptide of ER; Pin2-T, terminator of potato Pin2 gene; cDNA of the light chain (LC) and heavy chain(HC) of the mAb 13F6 in Figure 1A, respectively; E/35S-P, enhanced duplicated 35S promoter of Cauliflower mosaic virus; TEV, untranslated leader sequence of RNA4 of an Alfalfa mosaic virus; K, KDEL endoplasmic reticulum (ER) retention motif; NOS-T, terminator of nopaline synthase (NOS) gene; VL, variable region of LC of mAb 13F6; L, peptide linker. Expected protein structures of mAb 13F6-FULL and mAb 13F6-LSC. The symbols of the glycan structures are as follows: GlcNAc, black square; mannose, white circle. (**B**) Schematic diagram of Agrobacterium-mediated transformation to generate transgenic plant and mass production of transgenic plants expressing the 13F6 antibodies. A. tumefaciens strain LBA4404 carrying the plant binary vector was used for plant transformation. Agrobacterium was inoculated into the wounded plant leaf slices. Agrobacterium-inoculated leaf-slices were regenerated in 3–4 weeks. A few weeks later, small shoots were formed from the callus. These small shoots were transferred to the in vitro plant culture box. For biomass production, the in vitro transgenic plants were transferred into the soil pot and grown in a greenhouse.

**Figure 2 ijms-21-07007-f002:**
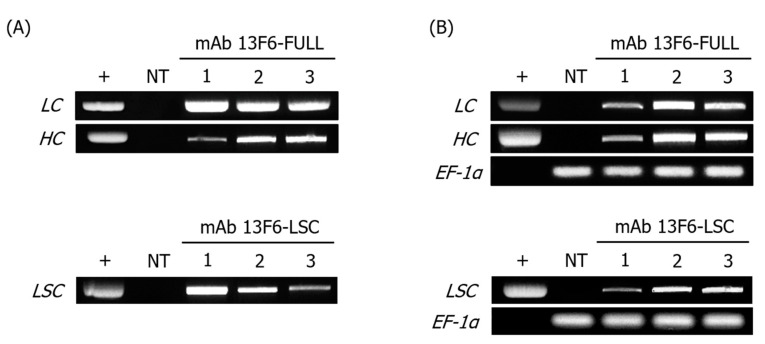
PCR and RT-PCR analyses of transgenic LC, HC, and LSC in gDNA and mRNA of the transgenic plant expressing mAbs 13F6-FULL and 13F6-LSC. (**A**) PCR analysis was performed to confirm the existence of LC and HC of mAb 13F6-FULL and LSC of mAb 13F6-LSC gene construct in the plant transformants. (**B**) RT-PCR analysis was conducted to confirm the mRNA transcription of LC and HC of mAb 13F6-FULL and LSC of mAb 13F6-LSC gene construct in the plant transformants. The relative transcription level of each gene was normalized by using the *EF-1α* gene as the internal control. +: positive control, pBIN 13F6-FULL and pBI 13F6-LSC in DH5α; NT: negative control, non-transgenic *N. tabacum* plant; 1–3: transgenic line #. The volume of sample loaded was 10 μL for each sample.

**Figure 3 ijms-21-07007-f003:**
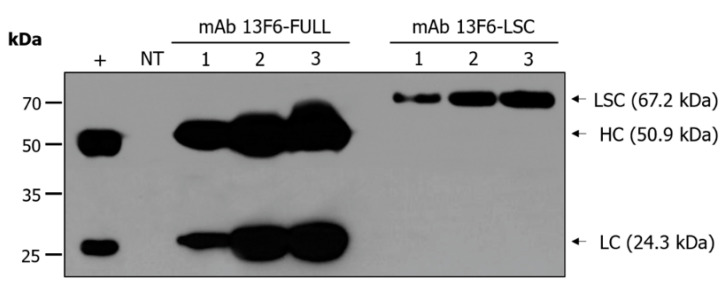
Immunoblot analysis of mAb 13F6-FULL and mAb 13F6-LSC in transgenic *N. tabacum* plants. An immunoblot analysis was performed to identify the LC and HC of mAb 13F6-FULL and LSC of mAb 13F6-LSC proteins in transgenic plants. Samples of each transgenic *N. tabacum* plant were homogenized with 1× phosphate-buffer saline (PBS) to confirm the protein expression level. The mAb 13F6-FULL and mAb 13F6-LSC were detected by the goat anti-human Fc fragment specific antibody and the goat anti-human immunoglobulin G (IgG) Fab fragment specific antibody, respectively. Numbers on the left indicate molecular weight (kDa). +: commercial 13F6 antibody (mAb 13F6-C) as the positive control; NT: non-transgenic *N. tabacum* plant as the negative control; 1–3: transgenic line #.

**Figure 4 ijms-21-07007-f004:**
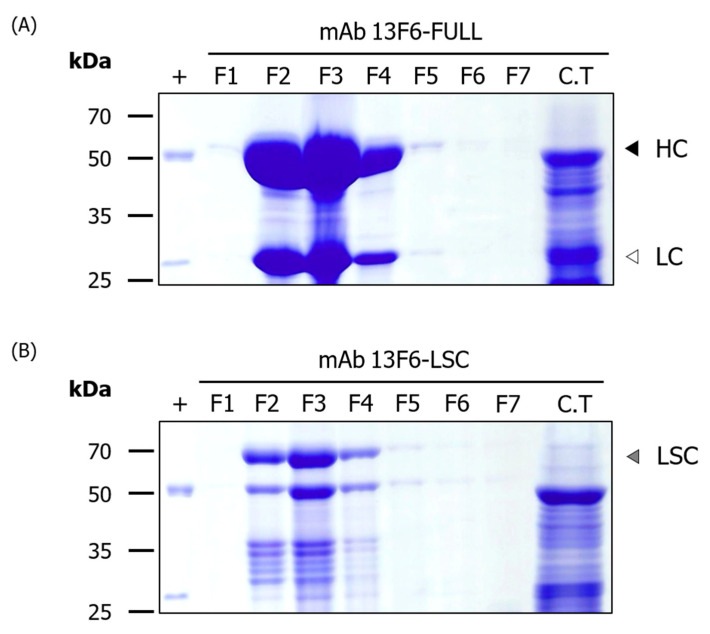
SDS-PAGE analysis to confirm the purification of mAb 13F6-FULL and mAb 13F6-LSC from transgenic N. tabacum plant. The eluted fractions of the purified mAb 13F6-FULL and mAb 13F6-LSC were visualized by Coomassie brilliant blue staining. Numbers on the left indicate molecular weight (kDa). +, commercial mAb 13F6 (mAb 13F6-C) as a positive control; F1–F7, purified samples from transgenic plants expressing mAb 13F6-FULL (**A**) and mAb 13F6-LSC (**B**); C.T, column through. Each well had 20 μL of the sample loaded in it. Black, white, and gray arrow heads indicate HC, LC, and LSC protein bands, respectively.

**Figure 5 ijms-21-07007-f005:**
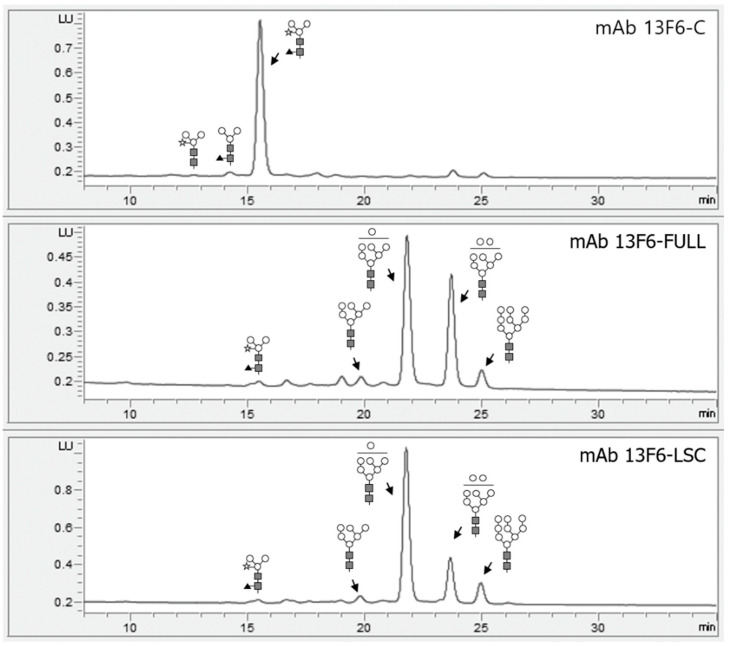
HPLC analysis of the *N*-glycosylation of mAb 13F6-C, mAb 13F6-FULL and mAb 13F6-LSC. The profiles of the 2-aminobenzamide (2-AB)-labeled *N*-glycan released from mAb 13F6-C (**top**), mAb 13F6-FULL (**middle**) and mAb 13F6-LSC (**bottom**) were analyzed using HPLC. The proposed *N*-glycan structures for each peak were designated. The symbols of the glycan structures are as follows: *N*-acetylglucosamine, gray square; mannose, white circle; xylose, white star; fucose, black triangle.

**Figure 6 ijms-21-07007-f006:**
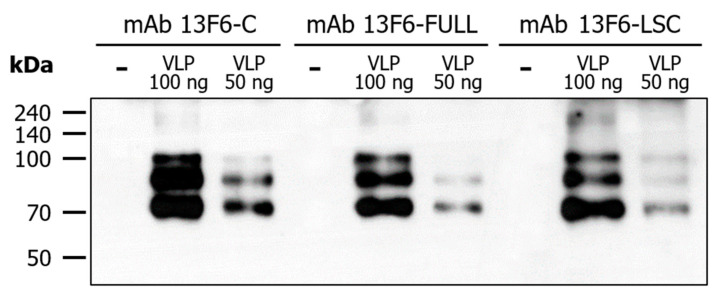
Detection of Ebola EBOV-like particles (VLPs) by mAb 13F6-FULL and mAb 13F6-LSC. Immunoblot analysis was performed to confirm the binding of mAbs 13F6 to VLPs. The VLPs were detected by mAb 13F6-C, mAb 13F6-FULL, or mAb 13F6-LSC and horseradish peroxidase (HRP)-conjugated goat anti-human IgG HC fragment specific antibody. Numbers on the left indicate molecular weight (kDa). −: negative control, 1× PBS; VLP: Ebola VLP; mAb 13F6-C, commercial 13F6 antibody as a positive control.

**Figure 7 ijms-21-07007-f007:**
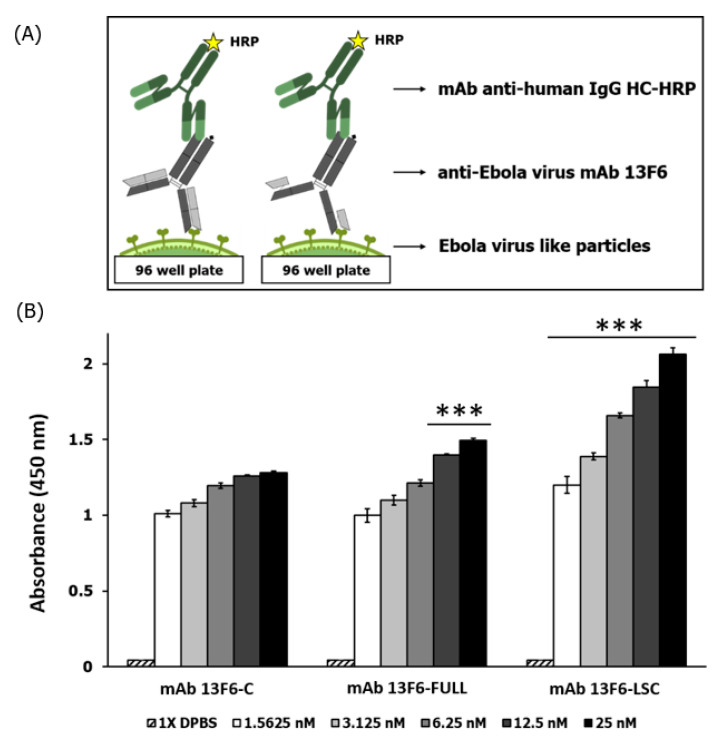
ELISA method to measure the binding affinity of mAb 13F6-FULL, and mAb 13F6-LSC to Ebola VLP. (**A**) Schematic diagram of an ELISA plate with mAb 13F6-FULL, and mAb 13F6-LSC to Ebola VLP. (**B**) Measurement of the binding activity of mAb 13F6-FULL and mAb 13F6-LSC to the Ebola VLP by indirect ELISA. The VLP (1 μg/well) was coated on the 96-well MaxiSorp immune plate. The primary antibodies were the mAb 13F6-C, mAb 13F6-FULL, and the mAb 13F6-LSC; the secondary antibody was HRP-conjugated goat anti-human IgG HC fragment antibody. Optical density levels of the binding affinity are presented as the mean ± standard deviation of three independent experiments. Asterisks indicate statistical significance by the Student’s *t*-test when compared with the 13F6-C control group (*** *p* < 0.001).

**Figure 8 ijms-21-07007-f008:**
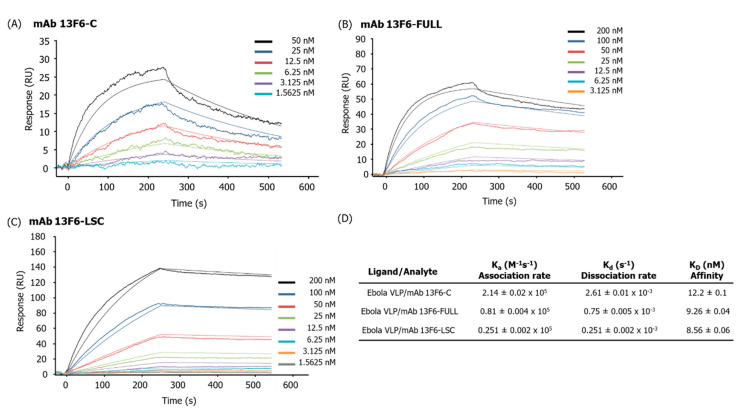
Surface plasmon resonance (SPR) analysis of the binding interactions of anti-Ebola virus (EBOV) mAb 13F6-C, mAb 13F6-FULL, and mAb 13F6-LSC to Ebola VLPs. (**A**) Two-fold serial dilution of the mAb 13F6-C (1.5625~50 nM) was injected onto the VLP surface. (**B**) Two-fold serial dilution of mAb 13F6-FULL (3.125~200 nM) was injected onto the VLP surface. (**C**) Two-fold serial dilution of mAb 13F6-LSC (1.5625~200 nM) was injected onto the VLP surface. SPR sensorgrams shown in thick lines were overlapped with the thin line and then fitted to a simple 1:1 interaction model curve. In each plot, the association and dissociation phases are seen. (**D**) The kinetic constant values are calculated using a simple 1:1 interaction model. The mAb 13F6-C, a commercial anti-EBOV antibody was injected as a positive control. VLP was fixed on an immobilized PEG sensor chip (approximately 1530 RU on the sensor chip).

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
