# Peer review of "Expression of a Large Single-Chain 13F6 Antibody with Binding Activity against Ebola Virus-Like Particles in a Plant System"

_ijms, 2020, doi:10.3390/ijms21197007_

Round 1
Reviewer 1 Report
The paper “Expression of a Large Single-Chain 13F6 Antibody with Binding Activity Against Ebola Virus Like Particles in a Plant System” is a simple, logical, clearly written manuscript, describing the development of new antibody formats for production in plant systems for diagnostics purposes. I have only few comments concerning grammar and format.
Title (and some other instances throughout text), “Virus Like”: Virus-like is better.
All over the paper: in some instances, species names and “in vitro” are not italicized.
All over the paper: you use the notations mAbP 13F6 and mAb 13F6 (with and without P). However, in several instances, it’s not clear at all why one is used over the other. As it’s pretty clear that only the commercial 13F6 is not produced on plants, I’d just drop this distinction, to make it easier to read.
Material and Methods in general: I’d be happy to have more precise references for the antibodies used in this study. For example, if I look for “AffiniPure Goat Anti-Human IgG, Fcγ fragment specific” at the Jackson ImmunoResearch website, I found 4 different references. Which one was used here?
Page 1, “on March 30, 2020 [6]”: this should be updated before publication.
Page 3, legend of Figure 1, “cDNA of the LC and HC of the mAb 13F6 in Figure 1(A), respectively”: I guess this reference to Figure 1(A) is wrong, as this IS Figure 1A.
Page 4, legend of Figure 1, “in vitro magenta box”: I see no magenta on the figure...
Figures 2, 5 and 7: there are Word paragraph marks inside the figures (next to the parentheses). This does not look professional...
Page 5/6/9, “Both the antibodies”, “all the three”, “both the purified”: the article is not necessary on this type of sentence.
Page 6, “Ebola VLPs (concentrations: 50 and 100 ng)”: this is rather amounts than concentrations, no?
Page 6, “The commercial anti-ZEBOV GP mAb 13F6 (mAb 13F6-C) and 1X PBS used as a positive and a negative control”: missing a WERE before USED.
Page 7, legend figure 7, “Expected schematic diagram”: no need to say EXPECTED.
Page 7, legend figure 7, “The primary antibody was the”: The primary antibodIES WERE.
Page 8, “different concentrations .... was applied”: WERE applied.
Page 8, legend figure 8, “SPR sensorgrams shown in thick lines were overlapped with the thin line was and ”: the WAS should be deleted here.
Page 9, “Moreover, an early treatment regime of patients with EVD can tremendously improve their chances of survival significantly”: you need to choose only one adverb to use in this sentence, either tremendously or significantly.
Page 10, “they displayed an efficient binding ability for Ebola VLPs that of the mAbP 13F6”: missing something on the sentence. Perhaps "displayed a MORE efficient binding ability for Ebola VLPs THAN that of"?
Page 10, “with the help of a peptide linker”: can you please specify the sequence of this linker?
Page 11, “The membranes were incubated with mAbP 13F6-FULL, mAbP 13F6-LSC, and the commercial anti-ZEBOV GP mAb 13F6”: can you please specify the precise amounts of recombinant antibodies used for this experiment (in mg/mL or any other unit like this? Dilutions are only useful when one knows the original Ab concentration, as it’s the case for the commercial antibodies).
Author Response
Thank you for considering our manuscript entitled “Expression of a Large Single-Chain 13F6 Antibody with Binding Activity Against Ebola Virus-Like Particles in a Plant System” toward publication in International Journal of Molecular Sciences. We are grateful to reviewers for the helpful comments and valuable suggestions for correction or modification. Here are the responses to the comments of reviewers. We very much hope the revised manuscript is accepted for publication in International Journal of Molecular Sciences.
Title (and some other instances throughout text), “Virus Like”: Virus-like is better.
-> Changed to “Virus-like” (Page 1 line 3)
All over the paper: in some instances, species names and “in vitro” are not italicized.
-> Those are now italicized.
All over the paper: you use the notations mAbP 13F6 and mAb 13F6 (with and without P). However, in several instances, it’s not clear at all why one is used over the other. As it’s pretty clear that only the commercial 13F6 is not produced on plants, I’d just drop this distinction, to make it easier to read.
-> In this study, indeed the commercial 13F6 is produced in plants. This information is described in M&M 4.4 SDS-PAGE and Immunoblot analysis (Page 12). To avoid the confusion, the fact that the commercial 13F6 is produced from plant has been mentioned through the manuscript. Also, ‘P’ has been removed from mAb through the manuscript.
Material and Methods in general: I’d be happy to have more precise references for the antibodies used in this study. For example, if I look for “AffiniPure Goat Anti-Human IgG, Fcγ fragment specific” at the Jackson ImmunoResearch website, I found 4 different references. Which one was used here?
-> The catalog # 109-035-008 has been included for the antibodies on Page12. (Page 12 line 407)
Page 1, “on March 30, 2020 [6]”: this should be updated before publication.
-> “on March 30, 2020” has been changed to “July 3, 2020”. Also, the digit information has been newly updated as follows; “the death of 2,299 people among 3,481 infected cases”. (Page 1 line 35-36)
Page 3, legend of Figure 1, “cDNA of the LC and HC of the mAb 13F6 in Figure 1(A), respectively”: I guess this reference to Figure 1(A) is wrong, as this IS Figure 1A.
-> It has been corrected to “Figure 1A”. (Page 4 line 121)
Page 4, legend of Figure 1, “in vitro magenta box”: I see no magenta on the figure...
-> “in vitro magenta box” has been changed to “in vitro plant culture box”. (Page 4 line 130-131)
Figures 2, 5 and 7: there are Word paragraph marks inside the figures (next to the parentheses). This does not look professional...
-> The word paragraph marks in Figures 2, 5, and 7 have been modified.
Page 5/6/9, “Both the antibodies”, “all the three”, “both the purified”: the article is not necessary on this type of sentence.
-> They have been changed to “Both antibodies”, “all three”, and “both purified”, respectively.
Page 6, “Ebola VLPs (concentrations: 50 and 100 ng)”: this is rather amounts than concentrations, no?
-> “concentrations” has been changed to “amounts”. (Page 6 line 186)
Page 6, “The commercial anti-ZEBOV GP mAb 13F6 (mAb 13F6-C) and 1X PBS used as a positive and a negative control”: missing a WERE before USED.
-> The sentence has been changed to “The commercial anti-ZEBOV GP mAb 13F6 (mAb 13F6-C) and 1X PBS were used as a positive and a negative control”. (Page 6 line 194)
Page 7, legend figure 7, “Expected schematic diagram”: no need to say EXPECTED.
-> The word “Expected” has been removed. (Page 7 line 209)
Page 7, legend figure 7, “The primary antibody was the”: The primary antibodIES WERE.
-> The words have been changed to “The primary antibodies were”. (Page 8 line 212)
Page 8, “different concentrations .... was applied”: WERE applied.
-> “was” has been corrected to “were”. (Page 8 line 222)
Page 8, legend figure 8, “SPR sensorgrams shown in thick lines were overlapped with the thin line was and ”: the WAS should be deleted here.
-> The words “was” has been removed. (Page 8 line 237)
Page 9, “Moreover, an early treatment regime of patients with EVD can tremendously improve their chances of survival significantly”: you need to choose only one adverb to use in this sentence, either tremendously or significantly.
-> The word “significantly” has been deleted. (Page 9 line 251)
Page 10, “they displayed an efficient binding ability for Ebola VLPs that of the mAbP 13F6”: missing something on the sentence. Perhaps "displayed a MORE efficient binding ability for Ebola VLPs THAN that of"?
-> The sentence has been corrected to “Moreover, they displayed a more efficient binding ability for Ebola VLPs than that of the commercial anti-ZEBOV GP mAb 13F6”. (Page 10 line 325-326)
Page 10, “with the help of a peptide linker”: can you please specify the sequence of this linker?
-> The sequence information (Gly-Gly-Gly-Gly-Ser)3 for the peptide linker has been included. (Page 11 line 366)
Page 11, “The membranes were incubated with mAbP 13F6-FULL, mAbP 13F6-LSC, and the commercial anti-ZEBOV GP mAb 13F6”: can you please specify the precise amounts of recombinant antibodies used for this experiment (in mg/mL or any other unit like this? Dilutions are only useful when one knows the original Ab concentration, as it’s the case for the commercial antibodies).
-> The precise information of the recombinant antibodies used in this experiment was 2 µg/ml. This information has been added to M&M 4.4 SDS-PAGE and immunoblot analysis. (Page 12 line 415-416)

Reviewer 2 Report
This is a strong methodology paper in which the authors present a method to produce Ebola antibodies in plants for diagnostics as a low cost option. The results compare a plant-isolated full antibody and a partial antibody construct including expression abundance, glycosylation patterns and binding affinity to the Ebola GP protein. The paper is very well written and easy to follow; the main issue to be addressed is to include more discussion of the results in the discussion section.
Introduction
Page 1 - perhaps use the word quarantine instead of isolate
Please include a few sentences why you chose to study the 13F6 antibody and not the other two antibodies found in the MB-003 therapeutic.
Please also briefly describe why you chose to test the LSC antibody in addition to the FULL antibody. Are there any benefits known to using LSC antibodies?
Results
Figure 2 - please provide figures of the entire gels as supplementary information
Figure 5 - is there a reason the glycan analysis was not also performed on the commercial 13F6 that has been used throughout the paper?
Discussion
Overall, the discussion is mostly just a summary of the results section and needs to be rewritten. For example, some of the discussion points could include:
- Figure 3 - the HC band is slightly higher than + ctrl band. what do you think this is? does this indicate a different glycosylation pattern?
- Figure 4 - it looks like you are losing a lot of H and C protein in the C.T. on the HC gel. would you agree? if so, why do you think this is?
- Why do you think the binding affinity of the plant produced Abs is higher than the commercially available one? Why is LSC affinity higher than HC?
The discussion is also a place to compare your findings to any similar or related publications. Are there other papers that have tried producing antibodies in plants? Did they find similar glycosylation patterns or differences between plant and mammalian antibodies?
Author Response
Thank you for considering our manuscript entitled “Expression of a Large Single-Chain 13F6 Antibody with Binding Activity Against Ebola Virus-Like Particles in a Plant System” toward publication in International Journal of Molecular Sciences. We are grateful to reviewers for the helpful comments and valuable suggestions for correction or modification. Here are the responses to the comments of reviewers. We very much hope the revised manuscript is accepted for publication in International Journal of Molecular Sciences.
Introduction
Page 1 - perhaps use the word quarantine instead of isolate
-> The word “isolate” has been corrected to “quarantine”. (Page 1 line 40)
Please include a few sentences why you chose to study the 13F6 antibody and not the other two antibodies found in the MB-003 therapeutic.
-> The sentence “In addition, the mAb 13F6 has several important characters such as a specific Mucin-like domain (MLD) binding activity without interfering interaction of mAb 12B5 to the MLD, which could provide important information for development of anti-EBOV mAb cocktails including advantage for quick EBOV detection (González-González et al., 2017).” has been added on Page 2 (Introduction). (Page 2 line 86-89)
Reference:
- González-González, E., Alvarez, M. M., Márquez-Ipiña, A. R., Trujillo-de Santiago, G., Rodríguez-Martínez, L. M., Annabi, N., & Khademhosseini, A. (2017). Anti-Ebola therapies based on monoclonal antibodies: current state and challenges ahead. Critical reviews in biotechnology, 37(1), 53–68. https://doi.org/10.3109/07388551.2015.1114465
Please also briefly describe why you chose to test the LSC antibody in addition to the FULL antibody. Are there any benefits known to using LSC antibodies?
-> The sentence “The large single-chain (LSC) antibody structure has several advantages. The light chain (LC) variable region and heavy chain (HC) coding sequences are genetically linked with the help of a peptide linker in a single transcript. Thus, there is no need to balance expressing proportion of the LC and HC. In addition, The LSC antibody required only one promoter to express in plant (Lee et al., 2020). This allows LSC antibody to be more quickly produced at lower costs than full size antibody (Bates & Power., 2019; Spadiut et al., 2014). The Fc region of LSC form facilitates efficient purification by affinity chromatography using protein G or protein A beads. Expression of two different targeting full size antibodies in a single plant could provoke misassembly of their own heavy and light chains, consequently resulting in chimeric antibodies with loss activities (Jamal et al., 2012).” has been included on Page 2 (Introduction). After the sentence “Expression of two different targeting full size antibodies in a single plant could provoke misassembly of their own heavy and light chains, consequently resulting in chimeric antibodies with loss activities (Jamal et al., 2012).”, the word “Thus,” has been added. In addition, the words “, which can be expressed in a single plant cell without the chimerism.” have been included after “a large single-chain (LSC) type antibody” on Page 2 (Introduction). (Page 2 line 73-83)
Reference:
- Lee, J.H., Park, S.R., Phoolcharoen, W. et al. Expression, function, and glycosylation of anti-colorectal cancer large single-chain antibody (LSC) in plant. Plant Biotechnol Rep 14, 363–371 (2020). https://doi.org/10.1007/s11816-020-00610-z
- Bates, A., & Power, C. A. (2019). David vs. Goliath: The Structure, Function, and Clinical Prospects of Antibody Fragments. Antibodies (Basel, Switzerland), 8(2), 28. https://doi.org/10.3390/antib8020028
- Spadiut, O., Capone, S., Krainer, F., Glieder, A., & Herwig, C. (2014). Microbials for the production of monoclonal antibodies and antibody fragments. Trends in Biotechnology, 32(1), 54-60. https://doi.org/10.1016/j.tibtech.2013.10.002
Results
Figure 2 - please provide figures of the entire gels as supplementary information
-> The figures of entire gels for Figure 2 were included as a supplementary Figure 1. Please see the entire gel photos in Figure S1. The sentence “Whole-gel images of Electrophoresis were provided in Supplementary Figure S1.” has been added to 2.1. Expression of mAb 13F6-FULL and mAb 13F6-LSC in transgenic N. tabacum plants. (Page 3 line 108)
Figure 5 - is there a reason the glycan analysis was not also performed on the commercial 13F6 that has been used throughout the paper?
-> The glycan analysis including the commercial 13F6 has been newly conducted. The glycan data of the commercial 13F6 have been included together with others as below. The “Result” and “Figure 5. Legend” parts for glycan analysis has been rewritten as follows;
2.3. N-glycan profile analysis of mAb 13F6-FULL and mAb 13F6-LSC using HPLC
HPLC was performed to analyze the N-glycan profiles released from mAb 13F-C, mAb 13F6-FULL and mAb 13F6-LSC (Fig. 5). mAb 13F6-FULL and mAb 13F6-LSC showed oligo-mannose-type glycan structures. They had a similar structure consisting of mostly 6-9 mannose glycans. mAb 13F6-FULL (Fig. 5 middle) had a significantly higher 8-mannose-type glycan structure than that of mAb 13F6-LSC (Fig. 5 bottom). However, in mAb 13F6-C, mainly N-glycans with β1,2‐linked xylose and α1,3‐linked fucose residues were observed without an oligo-mannose-type glycan structure (Fig. 5 top). (Page 6 line 170-177)
Figure 5. HPLC analysis of the N-glycosylation of mAb 13F6-C, mAb 13F6-FULL and mAb 13F6-LSC. The profiles of the 2-aminobenzamide (2-AB)-labeled N-glycan released from mAb 13F6-C (top), mAb 13F6-FULL (middle) and mAb 13F6-LSC (bottom) were analyzed using HPLC. The proposed N-glycan structures for each peak were designated. The symbols of the glycan structures are as follows: N-acetylglucosamine, gray square; mannose, white circle; xylose, white star; fucose, black triangle. (Page 6 line 179-183)
Discussion
Overall, the discussion is mostly just a summary of the results section and needs to be rewritten. For example, some of the discussion points could include:
- Figure 3 - the HC band is slightly higher than + ctrl band. what do you think this is? does this indicate a different glycosylation pattern?
-> According to reviewer’s suggestion, we have included the discussion sentences in ‘Discussion’ part as follows; From the immunoblot results in Figure 3, the HC and LC bands were slightly higher than that of the commercial 13F6 antibody. This is probably due to the difference in glycan structure as shown in figure 5. The glycosylation pattern of mAb 13F6-FULL is high mannose, whereas the glycosylation pattern of mAb 13F6-C was plant specific glycan structure with β1,2‐linked xylose and α1,3‐linked fucose residues. Thus, the molecular weight of mAb 13F6-FULL with high mannose glycan structure seems to have been shifted compared to the control 13F6-C. In previous studies, the molecular weight of recombinant GA733-2-Fc-KDEL proteins with high-mannose type N-glycans was higher than recombinant GA733-2-Fc with plant-derived N-glycan (Fu et al., 2018; Triguero et al., 2005). (Page 9 line 278-286)
Reference:
- Fu, Y. Y., Zhao, J., Park, J. H., Choi, G. W., Park, K. Y., Lee, Y. H., & Chung, I. S. (2018). Human colorectal cancer antigen GA733-2-Fc fused to endoplasmic reticulum retention motif KDEL enhances its immunotherapeutic effects. Journal of cancer research and therapeutics, 14(Supplement), S748–S757. https://doi.org/10.4103/0973-1482.199445
- Triguero, A., Cabrera, G., Cremata, J. A., Yuen, C. T., Wheeler, J., & Ramírez, N. I. (2005). Plant-derived mouse IgG monoclonal antibody fused to KDEL endoplasmic reticulum-retention signal is N-glycosylated homogeneously throughout the plant with mostly high-mannose-type N-glycans. Plant biotechnology journal, 3(4), 449–457. https://doi.org/10.1111/j.1467-7652.2005.00137.x
Figure 4 - it looks like you are losing a lot of H and C protein in the C.T. on the HC gel. would you agree? if so, why do you think this is?
-> The figure of Immunoblot analysis to confirm the purified fraction of mAb 13F6-FULL from transgenic N. tabacum plants for figure 4A was included as a Figure S2. As a result of immunoblot analysis confirming of the purified fraction and C.T of mAb 13F6-FULL, bands of HC and LC were not detected in C.T lane. In general, the bands, which look like HC and LC in the C.T lane from the SDS-PAGE result in Figure 4A, appear to be subunits of rubisco (d-ribulose 1,5-bisphosphate carboxylase/oxygenase) protein, which is the most abundant protein in plants (Bar-On & Milo., 2019). In previous studies, in the results of SDS-PAGE and western blot of LSC-type antibodies purified from plants with the protein A affinity column, the rubisco protein was confirmed at the position of 50 kDa (Lee et al., 2020). These sentences above have been included in “Discussion”. (Page 10 line 298-306)
Reference:
- Bar-On, Y. M., & Milo, R. (2019). The global mass and average rate of rubisco. Proceedings of the National Academy of Sciences of the United States of America, 116(10), 4738–4743. https://doi.org/10.1073/pnas.1816654116
- Lee, J.H., Park, S.R., Phoolcharoen, W. et al. Expression, function, and glycosylation of anti-colorectal cancer large single-chain antibody (LSC) in plant. Plant Biotechnol Rep 14, 363–371 (2020). https://doi.org/10.1007/s11816-020-00610-z
- Why do you think the binding affinity of the plant produced Abs is higher than the commercially available one? Why is LSC affinity higher than HC?
-> As I described above, the commercial one was also obtained from plant expression system. We authors believe that the higher affinity of LSC compared to the full sized mAb is due to the linker to fuse LC variable region (VL) to HC variable region (VH) which could give more flexibility for better affinity to the antigen. Thus, the sentences to discuss these points with references have been included in Discussion part (Page 10) as follows; In this study, the flexible glycine-rich-sequences as a linker was used to fuse VL to VH in mAb 13F6-LSC, which could give more flexibility for better affinity to the antigen than the commercial full size mAb (Gu et al., 2010). (Page 10 line 335-337)
Reference:
- Gu, X., Jia, X., Feng, J., Shen, B., Huang, Y., Geng, S., Sun, Y., Wang, Y., Li, Y., & Long, M. (2010). Molecular modeling and affinity determination of scFv antibody: proper linker peptide enhances its activity. Annals of biomedical engineering, 38(2), 537–549. https://doi.org/10.1007/s10439-009-9810-2
The discussion is also a place to compare your findings to any similar or related publications. Are there other papers that have tried producing antibodies in plants? Did they find similar glycosylation patterns or differences between plant and mammalian antibodies?
-> In this current study, the positive control, which is commercially available was plant-derived full size antibody. In this study, thus, we authors compared glycosylation profile among mAb 13F6-C (positive control), mAb 13F6-FULL, and mAb 13F6-LSC, which were all expressed in plant. According to suggestion, new sentences including discussion on previous studies producing antibodies in plants have been added as follows (Page 13 line 299); In previous studies, the anti-cancer antibodies with and without KDEL expressed in plant showed oligomannose type and plant-specific glycan profiles, respectively, whereas the antibodies expressed in mammalian cells had glycan structures harboring mammalian specific glycan resides, such as α1,6‐linked fucose, β1,2‐linked galactose, and α 2,6-sialic acid (Ko et al., 2005, Ko., 2014; Song et al., 2018; Lee et al., 2020). (Page 10 line 309-313)
Reference:
- Ko, K., & Koprowski, H. (2005). Plant biopharming of monoclonal antibodies. Virus research, 111(1), 93–100. https://doi.org/10.1016/j.virusres.2005.03.016
- Ko K. (2014). Expression of recombinant vaccines and antibodies in plants. Monoclonal antibodies in immunodiagnosis and immunotherapy, 33(3), 192–198. https://doi.org/10.1089/mab.2014.0049
- Lee, J.H., Park, S.R., Phoolcharoen, W. et al. Expression, function, and glycosylation of anti-colorectal cancer large single-chain antibody (LSC) in plant. Plant Biotechnol Rep 14, 363–371 (2020). https://doi.org/10.1007/s11816-020-00610-z
- Song, I., Kang, Y., Lee, Y. K., Myung, S. C., & Ko, K. (2018). Endoplasmic reticulum retention motif fused to recombinant anti-cancer monoclonal antibody (mAb) CO17-1A affects mAb expression and plant stress response. PloS one, 13(9), e0198978. https://doi.org/10.1371/journal.pone.0198978

Round 2
Reviewer 2 Report
Thank you for your responses and additional information.